# Patient-level Machine Unlearning in Latent Diffusion Models: On the Limits of the Privacy-Utility Trade-off

**Inês Cardoso**[1,2]
**Tiago Gonçalves**[2]
**Luís F. Teixeira**[2]
**Wilson Silva**[1]                                W.J.DOSSANTOSSILVA@UU.NL
[1] *AI Technology for Life, Department of Information and Computing Sciences, Department of Biology, Utrecht University, Utrecht, The Netherlands*
[2] *INESC TEC, Faculty of Engineering, University of Porto, Porto, Portugal*

## Abstract

While synthetic images are often used in federated learning for case-based explanations, diffusion models may introduce privacy risks by memorizing and potentially recreating identifiable data. Machine unlearning offers a way to remove specific training data influences, but its effectiveness for patient-level anonymization in generative models is not yet well understood. In this work, we present the first empirical analysis of patient-level unlearning in latent diffusion models, testing three strategies, including our novel KL-Away approach. Our results reveal a critical trade-off: methods that successfully unlearn data degrade diagnostic utility, whereas utility-preserving techniques fail to protect privacy, leaving over 20% of patients re-identifiable. We attribute this to feature entanglement and distributed memorization, suggesting that existing unlearning techniques are currently insufficient for reliable patient anonymization.

**Keywords:** case-based explanations, federated learning, machine unlearning, privacy.

## 1. Introduction

Case-based explanation systems justify artificial intelligence (AI) predictions by retrieving visually similar examples, aligning closely with clinical reasoning in radiology (Montenegro and Cardoso, 2025). In federated learning (FL) settings, where data cannot leave local institutions, such approaches require synthetic image catalogs as proxies for real patient data (Campos et al., 2024). Among generative approaches, diffusion models have emerged as a dominant paradigm for high-quality medical image synthesis (Croitoru et al., 2023). However, literature shows that these models memorize training data and reproduce patient-identifiable features in their outputs (Carlini et al., 2023), thus violating clinical privacy requirements and regulations such as the EU General Data Protection Regulation (GDPR), which mandates a Right to Be Forgotten (Dessers and Valcke, 2025).

Machine unlearning (MU) offers a potential solution by selectively removing the influence of specific training samples from a trained model without full retraining (Bourtoule et al., 2021). While MU has been studied for classification tasks and concept erasure in text-to-image models, its application to data-point unlearning in image-to-image diffusion models (particularly at the patient level in medical imaging) remains unexplored. Closest related works include SISS (Alberti et al., 2025), which targets individual entities one at a time, and

the approach proposed by Li et al. (2024), which degrades forget-set outputs to noise rather than preserving image coherence. Neither is designed for multi-patient anonymization.

We present a novel systematic study of patient-level MU in latent diffusion models for synthetic medical image generation. Using a re-identification pipeline, following the anonymization protocol defined by Packhäuser et al. (2022), we identify memorized patients, construct *Forget* and *Remain* sets, and introduce KL-Away, a Kullback-Leibler divergence-based unlearning method tailored to latent diffusion models. Through experiments on MIMIC-CXR-JPG (Johnson et al., 2019), we show that existing MU methods fail to reliably anonymize synthetic medical images, revealing a fundamental privacy-utility trade-off driven by feature entanglement and distributed memorization in latent space. The code for this work is publicly available[1].

## 2. Method

Given a latent diffusion model, we aim to remove patient-specific memorization such that synthetic outputs can no longer expose real individuals. Our pipeline has two stages: (1) identifying memorized patients and constructing the *Forget* and *Remain* sets; and (2) applying MU methods to suppress their influence on the model's output.

We generated 4,000 synthetic chest radiographs and applied a patient re-identification pipeline. For this purpose, we trained two ResNet-50 (He et al., 2016) Siamese networks on the real training data: a Patient Retrieval Network (contrastive loss) and a Patient Verification Network (classification loss). For each synthetic image, the retrieval network retrieves the three most similar real training images; the verification network then classifies whether the top match belongs to the same patient. Synthetic images with a verification score $\geq 50\%$ are flagged as non-anonymous.

Real images of patients associated with at least one flagged synthetic image form the *Forget* set $D_f$; the remaining patients form the *Remain* set $D_r$. This yields 885 non-anonymous synthetic images linked to 631 patients, leaving 3,115 images in $D_r$.

We evaluated three unlearning strategies—KL-Away, SISS, and SalUn+KL-Away—operating in the latent space of the Medfusion VAE (Müller-Franzes et al., 2023). We additionally tested **Gradient Ascent (GA)**, but it led to model collapse and is therefore excluded from quantitative comparison. Each method combines a forgetting loss on $D_f$ with the standard diffusion loss on $D_r$.

**KL-Away (proposed)** encourages the current model to diverge from a frozen copy of the original model $\theta_0$ on $D_f$. The loss is:

$$\mathcal{L}_{\text{KL-Away}} = -\frac{1}{2}\|\hat{\epsilon}_\theta(x_f, t, c_f) - \hat{\epsilon}_{\theta_0}(x_f, t, c_f)\|^2. \tag{1}$$

Under Gaussian diffusion assumptions, maximizing KL divergence between the current model and a frozen reference reduces to this squared difference. Unlike GA, KL-Away diverges from the model's own predictions, providing a more targeted forgetting signal.

**SISS** (Alberti et al., 2025) combines naive deletion with gradient ascent via importance sampling, balancing forgetting speed and model stability.

---

1. https://github.com/inescardos/KL-Away-Pipeline

Table 1: Evaluation before and after unlearning. ↑ higher is better, ↓ lower is better.

| Method | Non-Anon ↓ | UA (%) ↑ | ODA (%) ↑ | PIR (%) ↓ | AUC ↑ | FID ↓ |
|---|---|---|---|---|---|---|
| Original (no unlearning) | 885 | N/A | 77.9 | 15.8 | 0.88 | 53.5 |
| SISS | **525** | **81.3** | **86.9** | **6.8** | 0.52 | 179.2 |
| KL-Away | 602 | 78.9 | 84.9 | 7.2 | 0.83 | 84.9 |
| SalUn + KL-Away (50%) | 766 | 76.7 | 80.9 | 9.2 | **0.85** | **81.2** |

**SalUn+KL-Away** applies the SalUn saliency mask (Fan et al., 2024b) to restrict weight updates to parameters most responsible for memorization, combined with KL-Away.

## 3. Experiments

We use MIMIC-CXR-JPG (Johnson et al., 2024), restricted to postero-anterior views (15,223 images from 10,156 patients), with cardiomegaly as the target condition. After removing 222 mislabeled lateral-view images, we train a Medfusion latent diffusion model (Müller-Franzes et al., 2023) and generate 4,000 synthetic images with a balanced class split.

We report anonymization and unlearning metrics: number of non-anonymous images (Non-Anon), Unlearning Accuracy (UA), Overall Dataset Anonymization (ODA), and Patient Identifiability Ratio (PIR). Clinical utility is measured via area under the receiver operating characteristic curve (AUC) of a DenseNet-121 (Huang et al., 2017) cardiomegaly classifier trained on the synthetic data and evaluated on a real held-out test set. Image quality is assessed via Fréchet Inception Distance (FID) (Heusel et al., 2017).

Table 1 summarizes the results. SISS achieves the strongest anonymization (UA 81.3%, PIR 6.8%) but at a severe cost: AUC drops to 0.52 and FID to 179.2, rendering images clinically unusable. KL-Away offers a better balance (PIR 7.2%, AUC 0.83), while SalUn+KL-Away achieves the best utility–privacy compromise (PIR 9.2%, AUC 0.85). Full-parameter methods outperform localized variants, suggesting memorization is distributed across parameters. No method achieves complete anonymization. We attribute this to the high feature entanglement between $D_f$ and $D_r$ (76%), a configuration identified as worst-case for unlearning (Fan et al., 2024a), and the inability of a finite synthetic sample to capture all memorized patients, causing exposure of previously undetected identities.

## 4. Conclusion

We show that patient-level MU in latent diffusion models faces a fundamental privacy–utility trade-off that current gradient-based methods cannot resolve. High feature entanglement and distributed memorization in latent space make reliable patient-level forgetting challenging. Moreover, *Forget sets* built from finite synthetic samples cannot capture all memorized patients, potentially exposing previously undetected identities. Reliance on re-identification networks both to construct the *Forget* set and evaluate anonymization also introduces circularity; future work should explore complementary methods such as membership inference attacks. These findings suggest that reliable patient-level deletion requires not only new unlearning methods but also improved pipelines for identifying memorized patient PII. Until more robust methods emerge, disentanglement or differential privacy offer better trade-offs.

## Acknowledgments

This work was supported by the Dutch Research Council (NWO) through the AiNED XS Europa project NGF.1609.241.009.

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
