# OpenReview forum: "Patient-level Machine Unlearning in Latent Diffusion Models: On the Limits of the Privacy-Utility Trade-off"
_MIDL.io/2026/Short_Papers — MIDL 2026 - Short Papers Poster_

### Official Review · Reviewer_1WUf · 2026-04-28
**Review: Patient-level Machine Unlearning in Latent Diffusion Models: On the Limits of the Privacy-Utility Trade-off**

**Rating:** 3
**Confidence:** 5

**Review:**

The paper proposes KL-Away, an unlearning method tailored towards diffusion models. It works using a post-hoc filter that identifies memorized images according to re-identification models. For those images, KL-Away fine-tunes the model to produce different predictions from the original model. The paper shows that this method preserves downstream performance better than previous methods; however, it is less successful than previous methods at unlearning patients. The paper attributes this to a privacy-utility trade-off that requires further investigation.

The paper addresses an interesting problem. Quantitatively, the proposed method shows strong retention of downstream performance on real test data. However, this comes at the cost of having the highest number of remaining privacy issues after application. The method's main disadvantage is its high reliance on the re-identification model. The paper does not demonstrate its merit over using the re-identification model as a simple filter, rather than a more complex unlearning method that, even after application, retains a significant number of privacy issues. Furthermore, the training objective does not ensure that the method learns meaningful image representations. More sophisticated methods are required to ensure that image features related to privacy are unlearned while the general structure and quality of the images is retained, in order to justify unlearning steps after diffusion model training.

**Summary:**

The paper proposes KL-Away, an unlearning method tailored towards diffusion models. It works using a post-hoc filter that finds memorized images according to re-identification models. For those images, KL-Away finetunes the model to have a different prediction from the original model. The paper shows that this method preserves downstream performance better than previous methods, however, it is not as successful as previous methods in unlearning patients. The paper attributed this to a privacy-utility trade-off that requires further investigation.

**Strengths:**

- Important research problem. Machine unlearning provides safeguards for patients who are part of the training data of generative models.

- Clarity. The paper is clearly written and successfully guides the reader through the experiments.

- There is almost no performance loss for a notable improvement in terms of privacy.

**Weaknesses:**

- Comparison to baseline. The method does not compare itself to SalUN alone, which itself presents a method for class/concept unlearning. While this task is not entirely aligned, without this experiment it is unclear how much of the performance gain comes from KL-Away and how much comes from SalUN.

- The method relies on re-identification methods both for application and for validation. This poses a strong limitation, as it inevitably inherits the same disadvantages these methods already have, such as unreliability and lack of interpretability. Additionally, the paper does not mention related work in this area, such as Packhäuser et al. 2022, Dar et al. 2025, Dombrowski et al. 2025, Koeken et al. 2025.

- Methodologically, the method forces the entire model to produce images that are far from those of the original model. In theory, this should lead to images of poor quality, such as arbitrary noise, as the strength of KL-Away is increased. This trade-off is not discussed in the paper, and only a single run is shown.

- Unlearning should be compared to the downstream performance of simply leaving all Non-Anon images out of the downstream training, so basically using the re-id model as filter.

**Justification Of Rating:**

The paper presents an interesting problem that, at the current stage, lacks sufficient motivation. Currently, it teaches the generative model to produce arbitrary signals instead of the learned images. It should be shown that this is better than simply removing the memorized identities altogether before proceeding with a more detailed analysis of the privacy-utility trade-off. At this stage, I believe the method is rushed and has serveral disadvantages over existing methods. However, I also believe that this paper could profit a lot from on-site discussions which is why I am not recommending rejection.

---

### Decision · Program_Chairs · 2026-05-08

Accept (Poster)